# Dietary Preferences of Loggerhead Sea Turtles (*Caretta caretta*) in Two Mediterranean Feeding Grounds: Does Prey Selection Change with Habitat Use throughout Their Life Cycle?

**DOI:** 10.3390/ani13040654

**Published:** 2023-02-13

**Authors:** Giulia Mariani, Fabio Bellucci, Cristiano Cocumelli, Caterina Raso, Sandra Hochscheid, Chiara Roncari, Eliana Nerone, Sara Recchi, Federica Di Giacinto, Vincenzo Olivieri, Simone Pulsoni, Marco Matiddi, Cecilia Silvestri, Nicola Ferri, Ludovica Di Renzo

**Affiliations:** 1Faculty of Bioscience and Agro-Food and Environmental Technology, University of Teramo, 64100 Teramo, Italy; 2Istituto Zooprofilattico Sperimentale dell’Abruzzo e del Molise “G. Caporale”, 64100 Teramo, Italy; 3Ministry of Health, Directorate General for Animal Health and Veterinary Medicinal Products, Via G. Ribotta 5, 00144 Roma, Italy; 4Istituto Zooprofilattico Sperimentale del Lazio e della Toscana “M. Aleandri”, 00178 Roma, Italy; 5Marine Turtle Research Group, Department of Marine Animal Conservation and Public Engagement, Stazione Zoologica Anton Dohrn, Via Nuova Macello 16, 80055 Portici, Italy; 6Centro Studi Cetacei Onlus (CSC), 65125 Pescara, Italy; 7Italian National Institute for Environmental Protection and Research (ISPRA), 00144 Roma, Italy

**Keywords:** loggerhead sea turtles, *Caretta caretta*, Adriatic Sea, Tyrrhenian Sea, feeding ecology, dietary analysis

## Abstract

**Simple Summary:**

Loggerhead sea turtles have a flexible distribution from oceanic-pelagic to neritic-benthic areas. Their dietary preferences are expected to change from pelagic prey to benthic prey along with habitat use while they increase in size. This study evaluated the gastrointestinal contents of 150 loggerhead sea turtles stranded and/or bycaught along the Italian coasts of the central Adriatic Sea and central Tyrrhenian sea to assess the most common prey phyla. Information on the ingestion of litter has been collected for each carcass as well. This work highlights the need to deepen the knowledge of the feeding ecology and distribution of sea turtles at any developmental stage.

**Abstract:**

According to their life stage, the loggerhead sea turtle (*Caretta caretta*) is found in a wide range of habitats, from neritic to more oceanic areas. Their feeding habits are expected to change as they develop, along with habitat use. Juvenile sea turtles are hypothesized to feed on pelagic species in oceanic areas, shifting to more benthic prey during the subadult and adult stages. We analyzed the gastrointestinal content from 150 loggerhead sea turtles stranded and/or bycaught along the Adriatic coast of the Abruzzo and Molise regions (*n* = 89) and the Tyrrhenian coast of the Lazio and Campania regions (*n* = 61) from 2018 to 2021. Food items were identified to the lowest taxonomic level, and the frequency of occurrence was calculated for each taxon and most recurrent species to assess changes in prey selection during the development. The marine litter was categorized, and the frequency of occurrence was calculated for the ingestion of litter. The most recurrent taxonomic prey group recorded in the Adriatic sample was Arthropoda (94%), followed by Mollusca (63%) and Chordata (34%). In the Tyrrhenian sample, loggerhead sea turtles fed mostly on Mollusca (84%), Arthropoda (38%), and Chordata (26%). Surprisingly, the Adriatic-Tyrrhenian sample groups showed similar feeding behavior between juveniles, subadults, and adults. A similar correlation has been observed concerning the ingestion of litter. Moreover, this study confirms the opportunistic feeding behavior of loggerhead sea turtles and their high adaptability.

## 1. Introduction

The loggerhead sea turtle *Caretta caretta* (Linnaeus, 1758) is the most abundant sea turtle species in the Mediterranean Sea [1]. The Mediterranean subpopulation has been recently downlisted as “least concern” (LC) according to the IUCN Red List [2]. However, its status is highly dependent upon conservation projects. The spatial distribution of loggerhead sea turtles extends from oceanic to neritic areas and from pelagic to benthic habitats, including beaches where females nest. This elastic distribution makes this species subject to diverse anthropogenic threats that are highly concentrated in the Mediterranean Sea as a closed basin [3,4,5,6]. The anthropogenic pressure, pathologies [7,8,9], and sometimes their association may result in a consistent number of individuals being bycaught and stranded along the coasts. These stranding events represent an opportunity to gather data using multidisciplinary approaches to further our understanding of sea turtle ecology and distribution in the Mediterranean Sea for conservation. Among these, the dietary analysis of the food remains in the digestive tract can provide valuable information to correlate prey abundance with sea turtles’ distribution and habitat use.

Loggerhead sea turtles have a complex life cycle with characteristic ontogenetic habitat shifts [1,3,6,10]. After the hatchlings have reached the sea [11], the oceanic phase begins, and they head to the open sea where they spend most of their juvenile stage—with a curved carapace length (CCL) up to 59.9 cm [12,13]—feeding primarily on pelagic prey due to their limited diving capacity [10,14,15]. After the juvenile oceanic phase, a transitional phase to the subadult stage (CCL range, 60–69.9 cm) follows, where sea turtles use both oceanic and neritic habitats [16]. Sea turtles then start frequenting more benthic habitats, drawing closer to neritic areas, and their feeding strategy adapts [10]. Once they reach the adult stage and become sexually mature (CCL > 70 cm), they can be found in neritic areas, feeding mainly on benthic organisms [6,12,17,18,19]. However, Casale et al. [10] proposed a relaxed life history model for the Mediterranean subpopulation, as there is a lack of a specific oceanic phase. After an obligate epipelagic post-hatchling phase, turtles showed an amphi-habitat strategy [10], preferring to feed on benthic organisms such as mollusks and crustaceans [19,20,21]. In light of this dietary variability, to deepen our understanding of Mediterranean loggerhead sea turtles’ life history and related habitat use, there is a need for detailed and complete analyses of loggerhead sea turtles’ dietary preferences.

From an ecological perspective, the eastern and western Mediterranean Sea are essential foraging and nesting areas for loggerhead sea turtles. The Adriatic Sea (Figure A1) represents a critical foraging ground for this species at any developmental stage (hatchlings, juveniles, subadults, adults) [3,22]. The northern part of the basin has a higher biodiversity level due to the influence of freshwater from the Po river, which increases the level of nutrients. For this reason, this area is abundant in jellyfish, crustaceans, and mollusks, which are ideal prey for sea turtles [23]. At the southern end of the Adriatic Sea, the Gulf of Manfredonia (Puglia region), a broad, shallow area, represents another essential foraging ground, hosting individuals during the neritic stage [22]. The Tyrrhenian Sea provides important foraging grounds and nesting areas for loggerhead sea turtles. The Campania coasts play an important role due to the presence of benthic communities associated with seaweed [20,24,25].

Dietary analysis provides multiple types of information and allows observing individuals of different life stages and sex, giving an insight into their feeding preferences and habitat use, which is crucial in setting conservation strategies. Recently, different approaches have been adopted to study loggerhead sea turtles’ foraging ecology, such as stable isotopes [26,27], beak movement sensors [20], trophic diversity indexes [10], methods of points and numerical importance [28], and dietary DNA analysis [29]. However, the frequency of occurrence is the most widely used approach [10,16,19,28]. Dietary analysis also provides information about the frequency of litter ingested by sea turtles. A study by Tomas et al. [30] revealed that sea turtles in the Mediterranean Sea (Spanish waters) show low feeding discrimination, resulting in 75.9% of litter ingestion in necropsied carcasses. More recent studies conducted in the western Mediterranean Sea and the central Mediterranean Sea (Italian Waters) showed that the ingestion of litter has a frequency of 85%, and the most common plastic items are sheet-like (82%), fragmented (58.7%), and thread-like (25.6%) debris [31,32].

This study aimed to evaluate the dietary preferences of loggerhead sea turtles stranded and/or bycaught along the coasts of the Abruzzo, Molise, Lazio, and Campania regions (Figure A1) over four years (2018 to 2021) to better understand if, and in what measure, their feeding strategy changes during the development and along with their use of the sea. Additionally, the feeding ecology between two different feeding grounds of the Mediterranean Sea, the Adriatic and the Tyrrhenian was compared.

## 2. Materials and Methods

### 2.1. Sample Collection

Sea turtles stranded and/or bycaught along the Adriatic coasts of Abruzzo and Molise regions were collected from 2018 to 2021 and examined by the Abruzzo and Molise Regional Stranding Network (DG 2014 21/167 and DCA n.67 of 22/05/2019) [33,34]. The necroscopic examination was performed at Istituto Zooprofilattico Sperimentale dell’Abruzzo e del Molise (IZS-Teramo). Similarly, stranded/bycaught sea turtles from the Tyrrhenian were collected by the regional stranding network in Lazio (Tartalazio) and Campania from 2018 to 2021. Necropsies were carried out at Istituto Zooprofilattico Sperimentale del Lazio e della Toscana for turtles found along the coast of Lazio and at the Istituto Zooprofilattico Sperimentale del Mezzogiorno at Portici for turtles found along the coast of Campania. The contents from the esophagus, stomach, and intestine from each carcass were then sampled [35].

A database was created for Adriatic and Tyrrhenian samples, gathering information collected during the rescue and the necroscopic examinations, such as the date and location of the stranding/capture and sex. The life stage of the individuals was assessed considering the curved carapace length (CCL): individuals were identified as juveniles with a CCL ≤ 59.9 cm; subadults included CCL between 60 and 69.9 cm; and adults were with a CCL ≥ 70 cm [12,13,17,18,36].

### 2.2. Digestive Tract Content Analysis

The esophagus, stomach, and intestine contents isolated during the necropsy were rinsed with fresh water, sieved with a 1 mm mesh, and then divided following the protocol by Matiddi et al. [35]. The natural food (FOO) samples were analyzed as well. With the aid of the stereomicroscope and the taxonomic keys [37,38], the prey items were classified to the lowest taxonomic level. Taxonomic diversity in the dietary habits was assessed on multiple levels: (1) considering only the phylum (Chordata, Mollusca, Arthropoda, Tentaculata, Echinodermata, Plantae, Algae, and Annelida); (2) considering the taxonomic groups (Fish, Gastropoda, Bivalvia, Cephalopoda, Ascidiacea, Crustacea, Bryozoa, Plant, Algae, Annelida, Echinodermata, Scaphopoda, Cnidaria, and Polychaetes). Comparing the two datasets gave insights into the differences between foraging grounds.

No volume parameters were noted when dealing with dry contents. However, considering the long digestion activity of this species, the long-lasting transit of hard-bodied items compared to soft-bodied prey and the fragmentation of food during ingestion, digestion, and protocol workflow, this method of sampling could provide biased information. In any case, biomasses are not strictly essential for the aim of this study, and the hard-bodied items collected can be considered satisfying for the final discussion.

The litter was classified into different categories [35]: sheet-like material (SHE), fragments of hard plastic items (FRA), thread-like material (THR), industrial plastic (INDAPLA), foam (FOA), other plastic (POTH), and litter other than plastic (OTHER).

### 2.3. Statistical Analysis

To assess the most common prey in the Adriatic and Tyrrhenian seas, the frequency of occurrence (FO) was calculated as the percentage of individuals (considering the entire GI tract) where the phylum or taxonomic group was present. Among the most reported phyla, the frequency of occurrence was also calculated for the most recurrent species. A Fisher’s exact test was performed with R-Studio [39] to study (1) if there were statistical differences in the frequency of the phyla between the two areas considered, (2) to assess the relationship between the taxa and the groups of individuals in terms of life stage (juvenile, subadult, adult) for the Adriatic Sea and Tyrrhenian Sea, and also (3) considering the overall sample size. The most common prey taxa (arthropods, mollusks, fish) were considered to evaluate interdependence between life stage, sex, and feeding behavior, as they were represented in every group (male juveniles, female juveniles, male subadults, female subadults, male adults, female adults). A post hoc test (function *pairwise.fisher*) was performed where statistically significant differences were found.

The general frequency of litter ingestion was calculated for both sampled areas (Adriatic and Tyrrhenian seas) as the percentage of individuals that ingested litter. The FO per litter category was also calculated for both samples, considering each life stage as well, to obtain an estimation of the most frequent types of litter ingested. To investigate if there is a difference in the rate of ingestion between the life stages (juvenile, subadult, adult), a Fisher’s exact test was performed considering the two feeding grounds separately and also altogether.

By convention, a *p*-value of 0.05 was used to reject or accept the hypothesis.

## 3. Results

### 3.1. Sample Collection and Data

In this study, 150 sea turtles were evaluated: 89 individuals from the Adriatic Sea (65 identified as females and 24 as males) and 61 from the Tyrrhenian Sea (33 identified as females, 19 as males, and 9 remained unidentified). Among them, 12 sea turtles were bycaught (8 in the Adriatic Sea, 4 in the Tyrrhenian Sea), 136 were found stranded (81 in the Adriatic Sea, 55 in the Tyrrhenian Sea), and 2 sea turtles from the Tyrrhenian Sea died at the recovery center. The sea turtles’ CCL ranged from 19.2 cm to 107 cm for the Adriatic sample (CCL 58.34 ± 1.73 cm (mean ± SE)), while for the Tyrrhenian sample, the CCL ranged from 22 cm to 81.8 cm (average 58.20 ± 1.97 cm (mean ± SE)) (Figure 1). The life stage was assessed according to the CCL measurement. Among the Adriatic sample, 42 were defined as juveniles, 23 were subadults, and 24 were adults. In the Tyrrhenian sample, 27 were identified as juveniles, 15 as subadults, and 18 as adults, but 1 remained unidentified.

### 3.2. Digestive Tract Content Analysis

Among the individuals from the Adriatic Sea, the most frequent phyla recorded were Arthropoda (94.4%), Mollusca (62.9%), and Chordata, represented mainly by fish (34.7%) (Figure 2). In both Adriatic and Tyrrhenian samples, it was unfeasible to identify all the species of fish because only fishbones were found as food remains. Similarly, among the life stage-specific groups (juveniles, subadults, adults), in juveniles and subadults, Arthropoda had the highest FO%, followed by Mollusca and Chordata, while among the adults, Mollusca had the highest FO% (Figure 3, Table 1). The Arthropoda phylum was represented by crustaceans only. Among sea turtles that ingested arthropods (*n* = 84), most of the records were identified as crabs (100%) and hermit crabs (15.49%), except for 23 records of *Squilla mantis* (40.0%) and three records of *Balanidae*. Crustaceans were reported in every gastrointestinal sample from juveniles (100%), in the majority of subadults (95.8%), and in 78.3% of adults (Figure 2). The most recurrent species of crabs identified were the *Liocarcinus* sp. (69.1%)*, Goneplax rhomboides* (29.8%), and *Ilia nucleus* (14.3%) (Figure 4). The Gastropoda class (73.2%) and Bivalvia (62.5%) were the most represented classes of Mollusca. The species *Mytilus galloprovincialis* (37.1%) was recorded in two adult females (Figure 4): one fed only on blue mussels (total dry weight 384.2 g), and one fed on fish (dry weight 36.3 g) and blue mussels (dry weight 215.2 g) (Figure 5). Two alien species already reported in the Adriatic Sea have been observed among the sample from the Adriatic Sea: the blue crab *Callinectes sapidus* and the bryozoan *Tricellaria inopinata*.

Similar results were observed in the Tyrrhenian Sea, as Mollusca (83.6%), Arthropoda (37.7%), and Chordata represented by fish and seahorses (26.2%) (Figure 2 and Figure 6) were the most recorded phyla, with Mollusca showing the highest FO% in every group of individuals (juveniles, subadults, adults) (Table 2). The most recurrent classes of Mollusca were Bivalvia (64.7%) and Gastropoda (47.1%), but a few reports of Cephalopoda were recorded and identified as cuttlefish bones and beaks (27.5%). The crustacean species identified from the Adriatic Sea sample were the same as those identified in the Tyrrhenian Sea (*Liocarcinus* sp., *Goneplax rhomboides*, *Ilia nucleus*), except for one record of *Maja crispata* found only in the Tyrrhenian sample. Additionally, the only species of hermit crab classified to the species level was *Dardanus arrosor*.

The presence of marine litter was significant in both Adriatic and Tyrrhenian samples (34.8% and 91.5%, respectively). In the Adriatic Sea, the most recurrent category of ingested litter is thread-like (THR) and sheet-like (SHE), with a frequency of 58.1% and FO 35.5%, respectively. In the Tyrrhenian Sea, the sheet-like (SHE) litter was the category with the highest frequency of 75.9%, followed by fragmented plastic items (FRA), with a frequency of 57.4% compared to 3.2% in the Adriatic samples; THR accounted for 50%. The frequency of litter ingestion among the different life stages (juvenile, subadult, and adult) was very similar (Table 1 and Table 2). The most recurrent litter categories ingested by juveniles in both feeding grounds were SHE and THR, as observed in the entire sample.

### 3.3. Statistical Analysis

According to Fisher’s exact test and post hoc test, the phyla Mollusca and Arthropoda were statistically different between the Adriatic and Tyrrhenian seas (*p*-values < 0.05), suggesting a preference for arthropods and mollusks, respectively. However, there was no significant difference (*p*-value > 0.5) in the frequency of the phyla recorded between the groups of sea turtles (juveniles, subadults, adults) neither from the Adriatic Sea nor for the ones from the Tyrrhenian Sea, nor considering the entire sample size (150 sea turtles) (*p*-value > 0.3). This shows very similar feeding behaviors between all sea turtle life stages. Moreover, considering only the most common phyla preyed (arthropods, mollusks, fish), there was no correlation between the prey and the groups of individuals considering the sex (female and male) and life stages neither in the Adriatic (*p*-value = 0.66), nor in the Tyrrhenian (*p*-value = 0.98) seas, nor considering the entire sample size (*p*-value = 0.8). Concerning the ingestion of litter, no differences in the litter ingestion between juveniles, subadults, and adults were recorded neither in the Adriatic nor Tyrrhenian seas (*p*-value > 0.5) nor considering the entire sample size (*p*-value = 0.64).

## 4. Discussion

The loggerhead sea turtles show predatory and opportunistic feeding behavior in both Adriatic and Tyrrhenian feeding grounds [13,20]. According to the frequency of occurrence, benthic organisms such as crabs, mantis shrimps, gastropods, and bivalves seem to be the preferred prey of loggerhead sea turtles in the Adriatic and Tyrrhenian seas. This evidence was already suggested by previous studies conducted in the same geographic regions [10,13,40,41]. On the one hand, the inability to completely digest hard-bodied food items such as crustacean exoskeletons and gastropod shells, along with their long digestion time, may result in a high frequency of occurrence of these taxa compared to soft-bodied items (such as cephalopods and jellyfish). On the other hand, the high availability of crustaceans in nature may also derive from fishing activity [10,42,43,44]. Crabs are considered scavengers; as such, they could be among the first animals colonizing dredged seabeds to feed on fishery discards. Furthermore, crabs can be discarded during fishing operations as bycatch, becoming easy prey for sea turtles. The significant reporting of the necrophagous gastropods of the genus *Nassarius* (58.5% in the Adriatic Sea, 33.3% *Nassaridae* family in the Tyrrhenian Sea) found in the gastrointestinal tract along with crabs and fish seems to corroborate this hypothesis. Trawling is the most common fishing method in the central Adriatic Sea and along the Lazio and Campania coasts [20,45]. Sea turtles may follow and retrace trawlers’ trails, which usually remain unchanged over the season, to feed on fishery discards and other organisms. This hypothesis should be further investigated by satellite telemetry and vessel tracking, as it may have consequences on the conservation status of this species.

Among the species of crustaceans identified, the most recorded in the Adriatic Sea belongs to the genus *Liocarcinus* (69.1%), endemic to the Mediterranean Sea and associated with sandy bottoms from 0 to 20 m in depth, characteristic habitats of the Adriatic Sea. This genus was also found among the samples from the Tyrrhenian Sea but with a lower frequency (8.70%). However, the Tyrrhenian samples were more fragmented, and crabs were mainly classified as “Unidentified crabs” to avoid mismatch. *Liocarcinus* genus was already reported as a common prey for loggerhead sea turtles in the central Tyrrhenian Sea by previous studies. Travaglini et al. recorded a frequency of occurrence of 72% for *Liocarcinus vernalis*, which is in line with our data from the Adriatic Sea [46]. Additionally, a study conducted along the Domitian littoral [20] showed how brachyuran crabs are highly preyed on by sea turtles (100%) and identified the species *Liocarcinus vernalis* and *Portunus hastatus*. Our results confirmed that loggerhead sea turtles play a role as specialized predators of *Liocarcinus* crabs [47,48]. Moreover, this represents the first record of *Liocarcinus* crabs in the gastrointestinal tract of loggerhead sea turtles in the Adriatic Sea. Records were also found of other crustaceans described in the literature as common prey of sea turtles as well: the crabs *Goneplax rhomboides, Ilia nucleus, Inachus* sp., and the hermit crab *Dardanus arrosor* [13,20].

Benthic mollusks were highly represented in this study (62.9% Adriatic Sea, 83.6% Tyrrhenian Sea), and the frequency of occurrence was in agreement with previous studies in the Tyrrhenian Sea reporting benthic mollusks as the second highest ranked prey (66%) [20]. Among them, *Mytilus galloprovincialis* represented the most recurrent species identified in the Adriatic samples (37.1%). Two adult females were observed to have fed abundantly on blue mussels. Considering a large number of similar-sized mussels, we believe these individuals fed from aquaculture farms or on pillars of the oil platforms largely present along the Italian coast of the Adriatic Sea [49]. This is supported by the fact that it is unlikely for sea turtles to find such a large amount of *M. galloprovincialis* with valves of the same size easily. This evidence represents how prey selection can change in response to anthropogenic activities, underlining the opportunistic feeding behavior of loggerhead sea turtles. Moreover, the reporting of sea turtles as bioturbators in mussels farms in the Adriatic Sea highlights the importance of further understanding the interaction with these animals.

Arthropods were mainly preyed on by sea turtles in the Adriatic Sea compared to the sea turtles from the Tyrrhenian Sea that preferred mollusks. This difference may result from the morphology of the sea bottom in the two areas. The Tyrrhenian Sea is deeper than the Adriatic Sea; its continental shelf is generally limited along the coast, and the sea bottom is irregularly interrupted by ridges and volcanic structures, where sandy habitats alternate with rocky ones. The Adriatic Sea is shallower and characterized by sandy and muddy bottoms [49]. The difference in the morphology of the two basins can be translated into diverse phyla and species identified in the GE tract of sea turtles. However, aquaculture farms (mussel farming particularly) and the type of fishing methods performed in both areas can also influence the abundance and concentration of a specific type of prey.

Several studies report that sea turtles are more likely to feed on slow-moving fish instead of fast-moving species unless dead as fishery discards [13]. In this study, identifying fish species was unfeasible due to the highly fragmented samples. Only fishbones were found, except for three *Hippocampus hippocampus* (slow-moving fish) already reported in the literature [13,41], and one record of sardine, a fast-moving fish that was likely a fishery discard [13].

Loggerhead sea turtles have complex life-history patterns and can be found in many habitats over their life cycle. Juveniles are believed to stay in the oceanic zone and gradually shift to the neritic zone during development. Consequently, their dietary preference is expected to change to more benthic organisms while increasing in size [3,6,10,14]. This is correlated with the enhanced diving capacity and the increased bite force once they develop into adults [16]. Theoretically, juveniles are hypothesized to feed on more pelagic prey since they undergo the oceanic stage far away from neritic areas. Nevertheless, according to our data, there are no differences in the feeding strategy between the life stages, and benthic taxa (such as crustaceans and mollusks) were the most preyed upon by juveniles, indicating the early recruitment of juveniles towards shallower and neritic areas. This was in agreement with previous studies conducted in the Mediterranean Sea [10,13,26]. Other studies from the western Mediterranean Sea also confirmed the lack of evidence of a size-related habitat shift, identifying an intermediate neritic transitional phase for juveniles [13,26]. A line of evidence also supports this hypothesis: it is unlikely for small turtles to find benthic prey in deep waters (>200 m) due to their limited diving capacity. The high availability of benthic prey for juveniles in the Adriatic Sea could also be explained by considering the bathymetric profiles [50]. The central Adriatic Sea has relatively shallow waters, with a maximum depth of 150 m within 60 km from the coast [50]. However, further investigation should be conducted to understand which areas are more frequented. It has to be noted that, even though from our data (Figure 2), there are some taxa only present in juveniles’ gastrointestinal content (Algae, Annelida, Bryozoa, Cnidaria, Echinodermata, and Plantae) in both the Adriatic and Tyrrhenian samples, the number of records is too low to be significant and to be defined as a dietary preference.

A lack of correlation between feeding strategy and sex was also observed, confirming what was recently demonstrated by Haywood et al. through the analysis of stable isotopes [26]. This again underlines the opportunistic and highly adaptive feeding behavior of loggerhead sea turtles and the heterogeneous and not life-stage-related habitat use. However, considering that the number of females is higher than males in our samples, further analyses focused on the interdependence of sex and feeding strategy should be conducted.

This study has allowed the first detection of the alien species *Callinectes sapidus* and *Tricellaria inopinata* in the GI content of loggerhead sea turtles. This finding highlights the opportunistic feeding behavior of loggerheads and the likely abundance of these two species in the Adriatic Sea [51,52,53]. Moreover, dietary analysis of loggerhead sea turtles can be used as a tool to assess potential new invasive species.

The presence of ingested litter documented in this study reveals that the rate of ingestion is similar and consistent between juveniles, subadults, and adults (*p*-value > 0.55). This may suggest that sea turtles in the Adriatic and Tyrrhenian seas are equally exposed to litter pollution at any developmental stage, resulting in a similar rate of litter ingestion. This evidence could confirm the hypothesis, also suggested by the food analysis, that juvenile, subadult, and adult sea turtles are distributed in the same habitats and that juveniles are more prone to come closer to neritic areas earlier, lacking a clear and strict oceanic phase. Whether the ingestion of litter in juveniles is the cause or the consequence of an early approach to coastal areas, this should be investigated thoroughly. The approach to coastal areas may result in sea turtles being more exposed to marine litter and, consequently, a higher rate of litter ingestion. Thus, the early approach and the high rate of litter ingestion could impair the health status of sea turtles.

## 5. Conclusions

This study highlights the use of dietary analysis, providing insight not only into the feeding ecology and habitat use of the loggerhead sea turtle *Caretta caretta* in the Adriatic and the Tyrrhenian seas but also into marine biodiversity and alien species distribution. Our data suggest that the subpopulation of loggerhead sea turtles from the Central Mediterranean Sea lacks a strict oceanic phase and clear habitat shift. Juveniles are hypothesized to come closer to neritic areas in the early stages, where they can find the availability of benthic prey such as mollusks and crustaceans. However, the early recruitment of juveniles towards coastal areas highlights their exposure to anthropogenic activities and pollution. Further studies on the correlation between the categories of litter, the type of food ingested, and the life stages should be conducted for conservation strategies. Loggerhead sea turtles from the Adriatic and the Tyrrhenian seas show opportunistic and variable foraging behavior, probably adapting their diet to fishing activities and the presence of aquaculture farms. However, this possible interaction should be investigated thoroughly. This adaptability can be the key to reconsidering their life cycle. This study expanded the data on the phyla and species preyed on by the loggerhead sea turtles in the Adriatic and Tyrrhenian seas. The results illustrated the role of sea turtle’s gastrointestinal content as a capsule of biodiversity and identified the presence of two alien species in the Central Adriatic Sea: the crab *Callinectes sapidus* and the bryozoan *Tricellaria inopinata*. This study also highlighted the necessity to redefine a life stage classification that varies widely in the literature based on sexual maturity and distribution.

## Figures and Tables

**Figure 1 animals-13-00654-f001:**
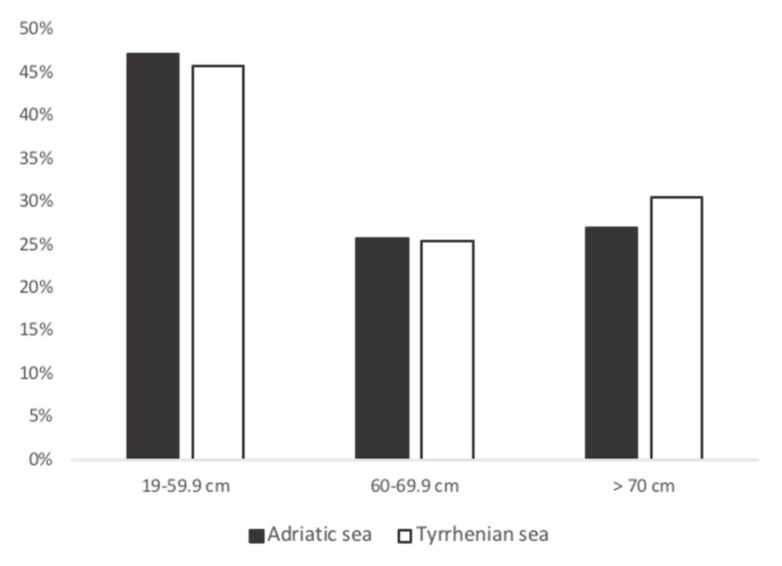
CCL frequency histogram in the Adriatic (*n* = 89) and Tyrrhenian seas (*n* = 61). A 19–59.9 cm CCL: *n* = 42 (Adriatic Sea), *n* = 27 (Tyrrhenian Sea); 60–69.9 cm CCL: *n* = 23 (Adriatic Sea), *n* = 15 (Tyrrhenian Sea); >70 cm CCL: *n* = 24 (Adriatic Sea), *n* = 18 (Tyrrhenian Sea).

**Figure 2 animals-13-00654-f002:**
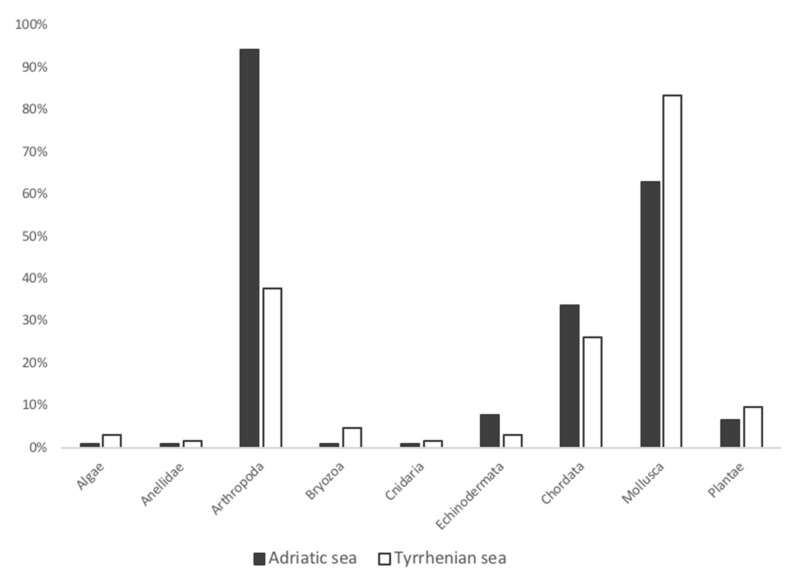
Frequency of occurrence of the phyla evaluated. Adriatic Sea FO%: Arthropoda, 94.4%; Mollusca, 62.9%; Chordata, 33.7%; Echinodermata, 8.9%; Plantae, 6.7%; Cnidaria, 1.1%; Bryozoa, 1.1%; Algae, 1.1%; Annelida, 1.1%. Tyrrhenian Sea FO%: Mollusca, 83.6%; Arthropoda, 37.7%; Chordata, 26.2%; Plantae, 9.8%; Bryozoa, 4.9%; Echinodermata, 3.3%; Algae, 3.3%; Cnidaria, 1.6%; Annelida, 1.6%.

**Figure 3 animals-13-00654-f003:**
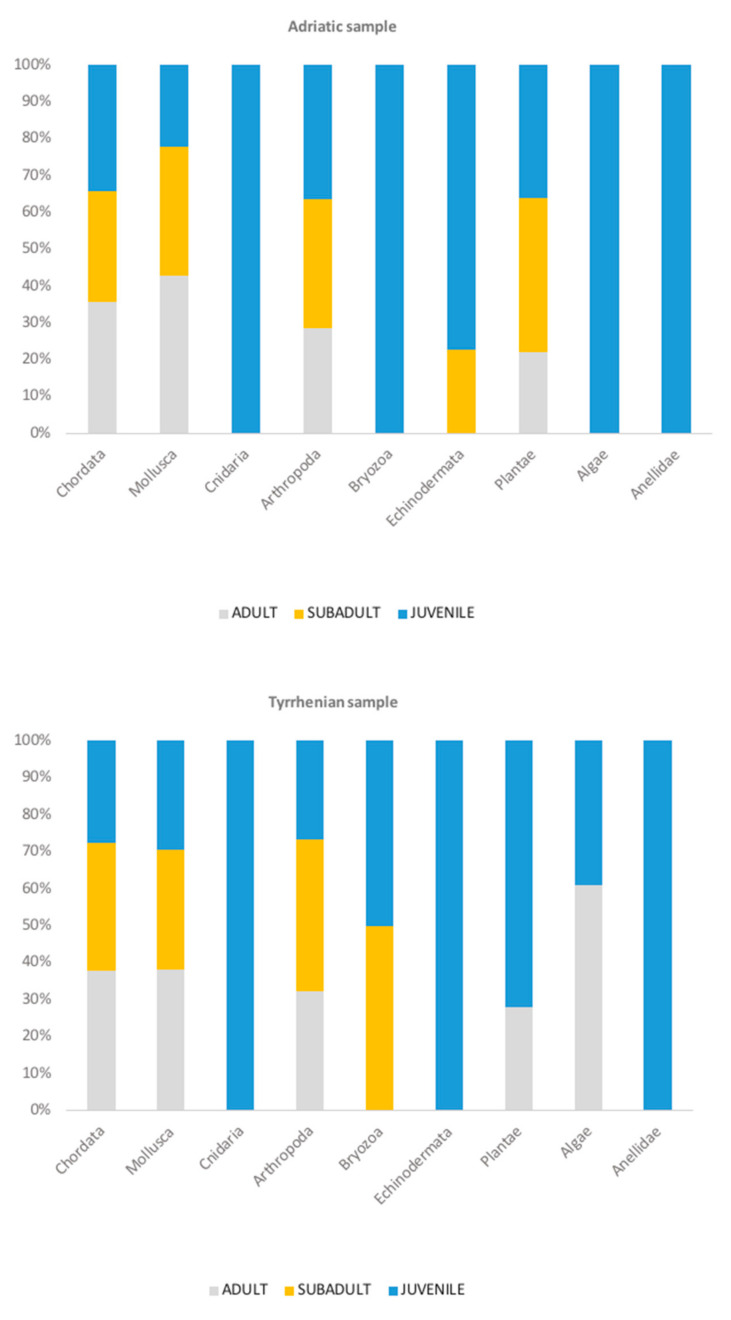
100% stacked bar chart: contribution of each life stage group per phyla of prey ingested.

**Figure 4 animals-13-00654-f004:**
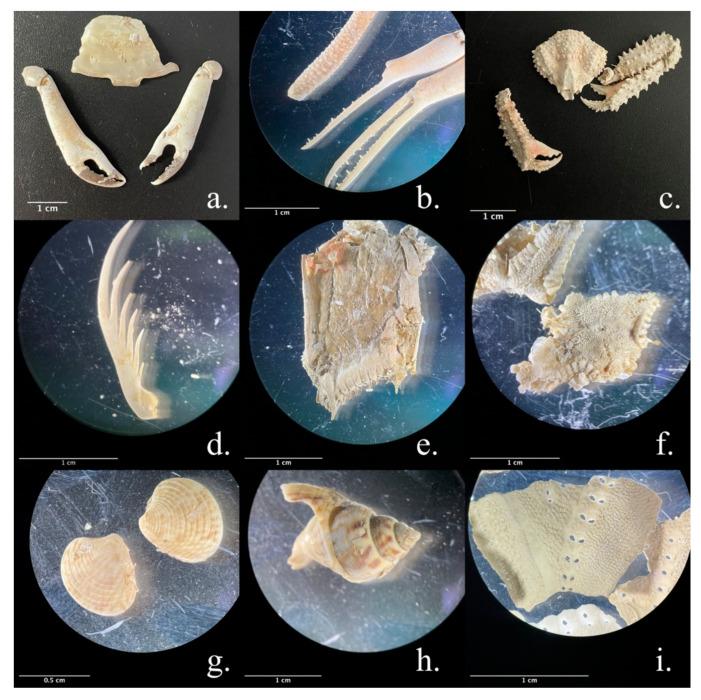
Multi-panel figure: items identified in the gastrointestinal content of sea turtles from the Adriatic Sea. (**a**) *Goneplax rhomboides*; (**b**) *Ilia nucleus*; (**c**) *Parthenope anguilifrons*; (**d**) *Squilla mantis*’ chela; (**e**) *Squilla mantis*’ tail; (**f**) starfish; (**g**) *Chamelea gallina* (Striped venus); (**h**) *Tritia mutabilis*; (**i**) Heart sea urchin.

**Figure 5 animals-13-00654-f005:**
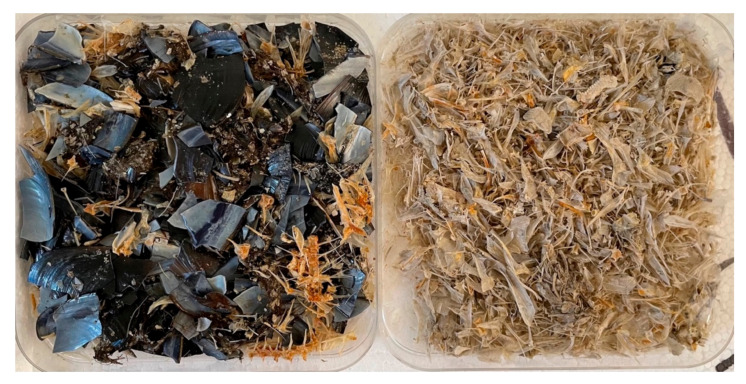
Gastrointestinal content of an adult female from the Adriatic Sea who predominantly fed on blue mussels and fish.

**Figure 6 animals-13-00654-f006:**
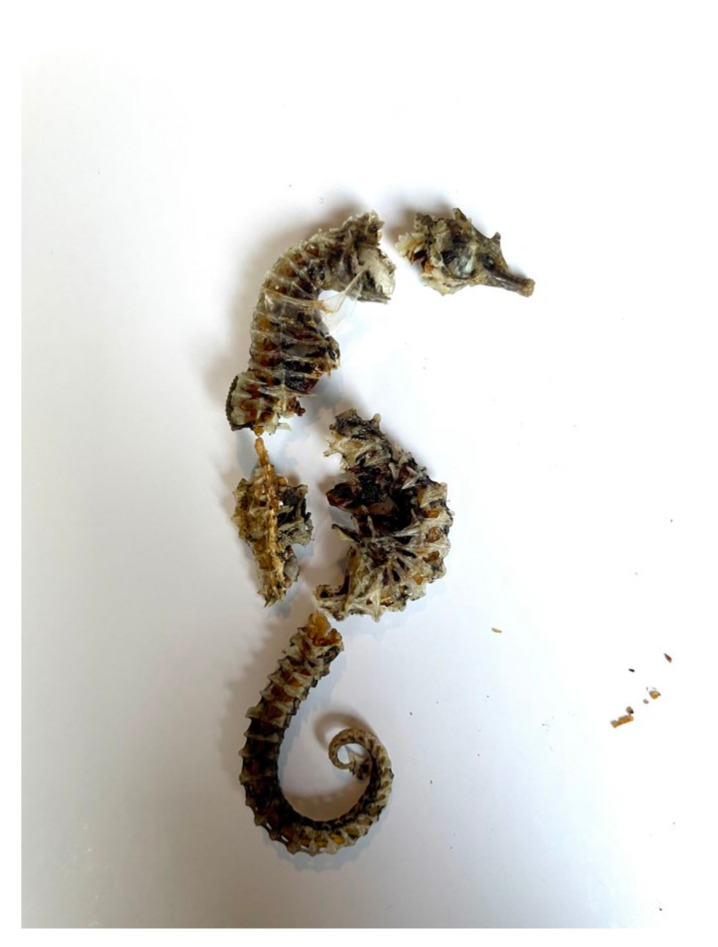
Individual of *Hippocampus hippocampus* found in one adult female from the Tyrrhenian Sea.

**Table 1 animals-13-00654-t001:** Frequency of occurrence of food and litter ingestion in loggerhead sea turtles from the Adriatic Sea: number of reports and FO% per phylum in the groups of juveniles (*n* = 42), subadults (*n* = 24), and adults (*n* = 23).

	Total		Juvenile		Subadult		Adult	
	*n* Reports	FO%	*n* Reports	FO%	*n* Reports	FO%	*n* Reports	FO%
Arthropoda	84	94.4	42	100	23	95.8	18	78.3
Mollusca	56	62.9	19	45.2	17	70.8	20	87.0
Chordata (fish)	30	33.7	14	33.3	7	29.2	8	34.8
Echinodermata	7	7.9	6	14.3	1	4.2	0	0
Plantae	6	6.7	3	7.1	2	8.3	1	4.4
Algae	1	1.1	1	2.4	0	0	0	0
Annelida	1	1.1	1	2.4	0	0	0	0
Bryozoa	1	1.1	1	2.4	0	0	0	0
Cnidaria	1	1.1	1	2.4	0	0	0	0
LITTER	31	34.8	15	35.7	8	33.3	8	34.8

**Table 2 animals-13-00654-t002:** Frequency of occurrence of food and litter ingestion in loggerhead sea turtles from the Tyrrhenian Sea: number of reports and FO% per phylum in the groups of juveniles (*n* = 27), subadults (*n* = 15), and adults (*n* = 18).

	Total		Juvenile		Subadult		Adult	
	*n* Reports	FO%	*n* Reports	FO%	*n* Reports	FO%	*n* Reports	FO%
Mollusca	51	83.6	18	66.7	10	66.7	15	83.3
Arthropoda	23	37.7	9	33.3	7	46.7	7	38.9
Chordata (fish)	16	26.2	8	29.6	5	33.3	7	38.7
Plantae	6	9.8	4	14.8	0	0	1	5.6
Bryozoa	3	4.9	2	7.4	1	6.7	0	0
Algae	2	3.3	1	3.7	0	0	1	5.6
Echinodermata	2	3.3	1	3.7	0	0	0	0
Annelida	1	1.6	1	3.7	0	0	0	0
Cnidaria	1	1.6	1	3.7	0	0	0	0
LITTER	54	91.5	26	96.3	13	86.7	13	72.2

## Data Availability

Not applicable.

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
