# Peer review of "Dietary Preferences of Loggerhead Sea Turtles (Caretta caretta) in Two Mediterranean Feeding Grounds: Does Prey Selection Change with Habitat Use throughout Their Life Cycle?"

_animals, 2023, doi:10.3390/ani13040654_

Round 1
Reviewer 1 Report
Nice job on this manuscript. I found it interesting and informative. I provided comments, questions and suggestions in an edited version of the manuscript. While I do recommend it for publication, I do think that revisions are required. I recommend that you include a section with the volume of species collected, similar to what you have for the %OF and compare the two sets of data. I would also like to see a bit more results and discussion on marine litter. I've given suggestions on the figures and even suggested additional figures that would give the readers more information. While you have a fairly large sample size, when you break it down into the three subcategories, juvenile, subadult and adult, your sample size becomes much smaller. I think that needs to be acknowledged. I also recommend evaluating how many of your samples could actually be oceanic stage verses those that have already settled into the neritic environment (see comments in manuscript). I know that will reduce your sample size even more.

Author Response
Dear Reviewer,
thank you for your preciouse comments and suggestions, please sea attachment for details in our response.
Best regards,
Authors

Reviewer 2 Report
The paper “Dietary preferences of loggerhead sea turtles (Caretta caretta) in two Mediterranean feeding grounds: is their opportunistic feeding strategy changing the habitat use during the lifecycle?” presents an interesting study on the diet of the Loggerhead turtle, with some insights on the ecologically important issues – litter consumption and presence of the invasive species. It is well structured and written. Still, it suffers from some minor inconsistencies in writing and could use more careful proofreading. I’ve pointed out some errors that I’ve found but the authors should, nevertheless, double-check the entire MS.
I think this study is a solid contribution to the field, but while reading it, I noticed a few things that the authors didn’t consider so I’ll make a few remarks:
1. There is a difference in the frequency of prey items consumed in the Adriatic and Tyrrhenian seas – the turtles in the Adriatic Sea seem to prefer arthropods, while the ones from the Tyrrhenian Sea prefer mollusks. Are these differences statistically significant (A simple Fisher’s test would suffice)? Also, how can they be explained keeping in mind that the Loggerheads are mostly opportunistic feeders – could it be influenced by the differences in the bottom configuration, differences in fishing and aquaculture, or general faunal differences in two seas? It could be and important discussion point.
2. What is the number of the stranded individuals, and the ones that came from the bycatch? Also, can stranding possibly be correlated with the litter ingestion (also check for literature on this)? For that, I’d test for differences in litter FO between stranded and bycatch individuals, to test a hypothesis that the stranded individuals had significantly higher litter consumption. It could also be a valid discussion point.
Other than that, I’d recommend this article to be published in Animals after a minor revision.
The further comments are in the annotated .pdf.

Author Response
Dear Reviewer,
thank you for your precious comments and suggestions, please sea attachment for details in our response.
Best regards,
Authors

Reviewer 3 Report
I think this reads as more of a report rather than a full length research article. There is a lack of references in the intro on strandings literature and authors should add some of the caveats of interpreting strandings data (location may not be where turtle was, but instead just where it was found, how to interpret "empty" stomaches, etc.). Line 93 is a very short treatment of references 17-23 and the authors do not adequately summarize this literature. What do those limited studies show (line 94, refs 3, 16)? Also missing from the Intro is other ways to explore diet such as electivity index (fish examples), species accumulation curves, in addition to FO.
There is a lack of hypotheses related to ocean currents - what would be expected vs. what was observed with stranded turtles? Was there anything interesting about where the turtles with high FO of fish were occurring (like near actively operating fisheries?)?
No study site figure - suggest to add one in
No significant findings (no significant differences in lines 249-261) so I would rewrite to be more of a story about similarities rather than differences. What about including diversity indices?
Same information seems to be presented in both tables and figures.
Some detailed edits required:
L 45 l missing from litter
L94 remove comma
L100 put in caveats of using strandings data, interpreting strandings data. add more intro material as to other diet studies in other regions that used stranded turtles to gain insight into diet of this species.
L120 small d in header - change to capital D
L205 and 207, these sentences are not making a paragraph. One sentence does not make a paragraph
3.3. No significant findings? Did you do any analyses by size, specifically? or location?
There are many suggestions for future work to be done sprinkled throughout - thus this is why my comment about this reads much more like a report rather than a peer reviewed full length scientific article.
L388 What is a capsule of biodiversity?
what was the condition of stranded turltes? did you only use ones in good condition?
Author Response

(The authors gave the same response as above.)

Reviewer 4 Report
Dear authors,
Many thanks for this piece of work. I read it with great interest and I found it meritorious of being considered further to publication.
I have just one comment regarding the conclusion. This section is definitely too long and 'wordy'. In the conclusion no room to arguments should be given but sharply head to the core of results, instead. So I would invite authors to shorten, focus and tailor the conclusion to main findings of this retrospective observation.
Thank you.
Author Response

(The authors gave the same response as above.)

Round 2
Reviewer 1 Report
Great job on the manuscript! It is obvious the authors took the reviewers comments into consideration and used them to improve the manuscript. I especially like the changes made to the figures. I have included some additional edits and suggestions in the attached file. I think with a bit of minor edits this manuscript is ready for publication.

Author Response
Dear Reviewer,
We thank you for your precious suggestions and help in improving the manuscript. We are pleased to know that you appreciate the improved manuscript.
Please find attached the detailed answers to your comments.
Best regards,
Authors

Reviewer 3 Report
Revised version of the manuscript is improved.
Author Response
Dear Reviewer,
We thank you for your comments and we are pleased that you find our manuscript improved.
Best regards,
Authors